# Biomarkers Related to Synaptic Dysfunction to Discriminate Alzheimer’s Disease from Other Neurological Disorders

**DOI:** 10.3390/ijms231810831

**Published:** 2022-09-16

**Authors:** Tommaso Piccoli, Valeria Blandino, Laura Maniscalco, Domenica Matranga, Fabiola Graziano, Fabrizio Guajana, Luisa Agnello, Bruna Lo Sasso, Caterina Maria Gambino, Rosaria Vincenza Giglio, Vincenzo La Bella, Marcello Ciaccio, Tiziana Colletti

**Affiliations:** 1Unit of Neurology, Department of Biomedicine, Neurosciences and Advanced Diagnostics, University of Palermo, 90129 Palermo, Italy; 2Department of Health Promotion, Mother and Child Care, Internal Medicine and Medical Specialties, University of Palermo, 90127 Palermo, Italy; 3Department of Biomedicine, Neurosciences and Advanced Diagnostics, Institute of Clinical Biochemistry, Clinical Molecular Medicine and Clinical Laboratory Medicine, University of Palermo, 90127 Palermo, Italy; 4Department of Laboratory Medicine, University Hospital “P. Giaccone”, 90127 Palermo, Italy; 5ALS Clinical Research Center and Laboratory of Neurochemistry, Department of Biomedicine, Neuroscience and Advanced Diagnostics, University of Palermo, 90129 Palermo, Italy

**Keywords:** Alzheimer’s disease, biomarkers, neurogranin, α-synuclein

## Abstract

Recently, the synaptic proteins neurogranin (Ng) and α-synuclein (α-Syn) have attracted scientific interest as potential biomarkers for synaptic dysfunction in neurodegenerative diseases. In this study, we measured the CSF Ng and α-Syn concentrations in patients affected by AD (*n* = 69), non-AD neurodegenerative disorders (*n*-AD = 50) and non-degenerative disorders (*n*-ND, *n* = 98). The concentrations of CSF Ng and α-Syn were significantly higher in AD than in *n*-AD and *n*-ND. Moreover, the Aβ42/Ng and Aβ42/α-Syn ratios showed statistically significant differences between groups and discriminated AD patients from *n*-AD patients, better than Ng or α-Syn alone. Regression analyses showed an association of higher Ng concentrations with MMSE < 24, pathological Aβ 42/40 ratios, pTau, tTau and the ApoEε4 genotype. Aβ 42/Ng was associated with MMSE < 24, an AD-related FDG-PET pattern, the ApoEε4 genotype, pathological Aβ 42 levels and Aβ 42/40 ratios, pTau, and tTau. Moreover, APO-Eε4 carriers showed higher Ng concentrations than non-carriers. Our results support the idea that the Aβ 42/Ng ratio is a reliable index of synaptic dysfunction/degeneration able to discriminate AD from other neurological conditions.

## 1. Introduction

Alzheimer’s disease (AD) is a severe neurological disorder, clinically characterized by a progressive loss of memory and cognitive impairment. Typical AD-related hallmarks are the extracellular deposition of plaques with β-amyloid peptides (i.e., Aβ 42) and the presence of intracellular neurofibrillary tangles containing the phosphorylated form of the Tau protein (pTau) [1,2].

To date, the strongest pathogenetic hypothesis has been the so-called amyloid cascade, whose first step is the overexpression of Aβ 42 oligomers and/or the failure of their clearance [3]. Aβ 42 aggregates and precipitates in senile plaques. They can have a toxic effect, resulting in the hyperphosphorylation of the Tau protein and the formation of neurofibrillary tangles. Tau is a microtubule-associated protein with a significant role in maintaining the integrity of neuronal cells, and its hyperphosphorylation causes a gain of function with a toxic effect on brain cells. Both processes (Aβ 42 and pTau deposition) interact with each other, leading to synaptic dysfunction and resulting in neuronal death. The timing of the amyloid cascade is reflected in the sequence of changes in CSF protein levels. The CSF level of Aβ 42 is reduced early in the disease (pre-clinical stages), while the CSF Tau levels (both pTau and tTau) increase as the result of their release after neuronal death. Aβ 42 and pTau are highly specific and are considered “AD-core biomarkers” for AD, whereas tTau is not, reflecting a neurodegenerative pathway that is common to other neurological disorders [2,4,5].

Neurodegeneration is the result of the aforementioned processes [6] and can be detected by CSF tTau measurement, conventional magnetic resonance imaging (MRI) able to detect brain atrophy, and fluorine-18-fluorodeoxyglucose positron emission tomography (FDG-PET), which is used to measure brain metabolism. FDG-PET is considered to reflect the spatial distribution of synaptic dysfunction, being associated with clinical features [7,8]. The typical AD hypometabolism spatial pattern is of temporo-parietal and posterior cingulate/precuneus, is highly specific, and predicts the progression of the disease [7,9,10,11,12,13]. FDG-PET, as well as CSF Aβ 42, pTau, and tTau, is widely considered to be a reliable biomarker for the diagnosis of AD and for predicting the progression of the prodromal stages of the disease to dementia [10,14]. However, the use of CSF biomarkers is not recommended as a routine procedure, but in selected cases or for research purposes [15,16]. The neuropathological and clinical heterogeneity of sporadic AD [17,18], and the common presence of comorbidities [19,20,21], as well as the alteration of typical CSF AD biomarkers in different conditions such as synucleopathies (including Parkinson’s disease dementia and dementia with Lewy bodies), cerebro-vascular diseases [22] and brain trauma [23], highlights the need for additional biomarkers able to improve the diagnostic accuracy in a real-world setting [24].

Recently, synaptic proteins have been considered valuable biomarkers for neurodegenerative diseases such as Parkinson’s disease (PD) and AD. Animal models have demonstrated that synaptic alterations precede neuronal loss in the pathogenetic pathway of AD [25]. Furthermore, synaptic loss is highly correlated with the level of cognitive impairment [26,27]. Both Aβ and Tau have a role in normal synaptic function, and their toxic forms (Aβ 42 and pTau) are involved in synaptic degeneration [28].

The identification of specific biomarkers related to synaptic integrity, able to rapidly diagnose patients with AD and predict progression from mild cognitive impairment (MCI) to dementia, can be helpful in the prodromal and preclinical stages, which are potential targets for the early treatment of the disease. Among these, the role of neurogranin (Ng) and α-synuclein (α-Syn) in AD-related synaptic dysfunction/disruption was recently discussed [29,30,31,32].

Ng is a postsynaptic protein of 78 amino acids, primarily expressed in the hippocampus and the pyramidal cells, which plays a crucial role in synaptic plasticity. This is a substrate for the protein kinase C (PKC), acting as a link between calmodulin and PKC signaling in synaptic plasticity. High levels of CSF Ng were detected in patients with AD and correlated with cognitive impairment, especially in the early stages of the typical AD phenotype, with prominent memory loss, as a result of early hippocampal degeneration [33,34,35,36]. Cross-sectional and longitudinal studies showed that CSF Ng levels have diagnostic and prognostic utility in AD patients even in the preclinical stages [37]. They can differentiate AD from other neurodegenerative diseases and healthy controls. However, their real specificity is still a matter of debate, because some authors have found higher levels of CSF Ng in neurodegenerative diseases other than AD, such as Creutzfeldt–Jakob disease (CJD) [38,39] and Parkinson’s disease (PD) [31,40].

α-Syn is a presynaptic protein of 140 amino acids, widely expressed in CNS neurons and localized in proximity to synaptic vesicles. Although its molecular functions are poorly defined, it plays a role in modulating synaptic activity, the density of synaptic vesicles and neuronal plasticity [41]. Certain neurodegenerative disorders, such as PD and dementia with Lewy bodies (DLB), can be related to mutations of the α-Syn gene (“synucleopathies”) and are characterized by the presence of pathological inclusions containing α-Syn filaments in cellular bodies and in the dendrites and axons of affected neurons [42,43]. Although not specifically, α-Syn is involved in the pathophysiology of AD. Some authors found high levels of CSF Ng in the early stages of disease, and their correlation with cognitive impairment [44,45,46] and α-Syn seems to interact with Aβ and Tau [47,48]. Moreover, Ng and α-Syn showed a positive correlation with pTau and tTau and negative correlation with Aβ 42 and the Aβ 42/40 ratio [29,49,50].

The apolipoprotein genotype E ε4 (APO E4) is known to be the strongest genetic risk factor for late-onset AD, but also for cognitive impairment in other neurodegenerative diseases, mainly for PD, in a dose-dependent way [51,52,53]. Recently, the interaction between APO E4 and CSF Ng was investigated in a population of AD and MCI patients, and the authors found higher levels of the protein in MCI carrying APOE e4(+) than APOE e4(−), suggesting an effect of APOE in determining the levels of CSF Ng in the early stages of AD [54]. The APOE genotype can also influence pathological changes, progression rates and cognitive impairments in animal models and patients with synucleopathies [53,55].

Recently, the role of the Aβ-42/Ng ratio was studied. It was first introduced to check whether the combination of the two biomarkers improved the differential diagnosis of AD dementia., as well as for the Aβ 42/40 ratio. Janelidzed and colleagues [56] showed that the diagnostic performance of the Aβ-42/Ng ratio did not differ from that of the Aβ 42/40 ratio, when they compared AD to non-AD dementias or cognitively stable MCI from MCI that converted to AD. Subsequently, the Aβ-42/Ng ratio has been demonstrated to be an index of synaptic impairment in PD, as Aβ physiologically regulates synaptic function and Aβ-42 is involved in synaptic toxicity [57]. Although no significant difference was found between the PD and control groups, the Aβ-42/Ng ratio showed a significant correlation with cognitive impairment in PD and is able to discriminate PD patients with cognitive deficits [58].

In the present study, we investigated the role of Ng and α-Syn in AD, as potential biomarkers of synaptic dysfunction. We compared patients affected by AD with patients affected by other neurodegenerative (*n*-AD) and non-neurodegenerative disorders (*n*-ND), to better define the role of synaptic dysfunction in their etiopathogenesis and in disease progression, as expressed in terms of cognitive impairment. Moreover, the Aβ-42/Ng ratio and Aβ-42/α-Syn were taken into account as potential indices of synaptic dysfunction in AD pathology.

## 2. Results

### 2.1. Participants’ Features

We analyzed data from sixty-nine (*n* = 69) AD patients, fifty (*n* = 50) *n*-AD patients, and ninety-eight (*n* = 98) *n*-ND patients. As shown in Table 1, the groups did not differ in gender and education, but AD and *n*-AD differed from *n*-ND in the age at time of lumbar puncture (LP) and cognitive status (MMSE). The list of *n*-AD and *n*-ND diagnoses are reported in Appendix A, respectively.

### 2.2. Differences in CSF Ng and α-Syn Levels between AD and n-ND Patients

As illustrated in Figure 1, we found significantly higher levels of CSF Ng (Figure 1a) and α-Syn (Figure 1b) in AD compared to *n*-ND patients (Ng: AD = 388 (238–531) pg/mL vs. *n*-ND = 242 (95–375) pg/mL, *p* < 0.001; α-Syn: AD = 2756 (2226–3164) pg/mL vs. *n*-ND = 1735 (1218–2393) pg/mL, *p* < 0.001), with a size effect that did not influence the final results (Ng: η^2^ = 0.095; α-Syn: η^2^ = 0.169).

The diagnostic performance of Ng and α-Syn for AD versus *n*-ND was evaluated by receiver operating characteristic (ROC) curve analysis (Figure 1c,d). The area under the curve (AUC) of Ng was 0.699 (C.I. 95% (0.611–0.787)), and the Youden’s cut-off was 165.5 pg/mL, with a sensitivity of 93% and a specificity of 41%. The AUC of α-Syn was 0.754 (C.I. 95% (0.672–0.835)), and the Youden’s cut-off was 1907.5 pg/mL, with a sensitivity of 89% and a specificity of 62% (Figure 1d).

### 2.3. Biomarkers Related to Synaptic Dysfunctions and AD-Core Biomarkers to Discriminate AD from n-AD

We then compared the CSF concentrations of synaptic-related proteins (i.e., Ng and α-Syn) and AD-core biomarkers (i.e., Aβ-42, pTau, tTau, and the Aβ 42/40 ratio) between AD and *n*-AD; we found, as expected, statistically significant differences in Aβ-42 (*p* = 0.017), the Aβ 42/40 ratio (*p* < 0.001), pTau (*p* < 0.001) and tTau (*p* < 0.001); the Aβ42 concentration and Aβ 42/40 ratio were lower in the AD group, while the pTau and tTau concentrations were higher (Appendix A). Interestingly, we also found higher CSF Ng and α-Syn in AD than *n*-AD (Ng: AD = 388 (238–531) pg/mL vs. *n*-AD = 192.5 (85.2–295) pg/mL, *p* < 0.001; α-Syn: AD = 2756 (2226–3164) pg/mL vs. *n*-AD = 2265 (1293–2542) pg/mL, *p* = 0.031), with a small size effect (Ng: η^2^ = 0.242; α-Syn: η^2^ = 0.398).

ROC analyses (Figure 2c,d) showed that the Ng levels were able to discriminate AD from *n*-AD patients (AUC = 0.768; C.I. 95% (0.682–0.853); *p* = 0.004), with a sensitivity of 68% and a specificity of 78%, while the AUC of α-Syn did not reach statistical significance (AUC = 0.689; C.I. 95% (0.514–0.804); *p* = 0.076).

We next investigated whether CSF Ng and α-Syn improved the differential diagnosis between AD and *n*-AD when compared to Aβ 42. Given the fact that the Aβ 42/40 ratio performs better than Aβ42, we assessed the difference in Aβ42/Ng and Aβ42/α-Syn and their ability to discriminate AD from *n*-AD. We found significantly lower values in AD than in *n*-AD (Aβ42/Ng, *p* < 0.001; Aβ 42/α–Syn = 0.001, respectively), similarly to what was seen for Aβ 42/40 (Table 2).

When we considered the ability to discriminate AD from *n*-AD by ROC curve analysis, we found an increase in diagnostic performance for both indices (Aβ 42/Ng: AUC = 0.814; C.I. 95% (0.684–0.944); *p* = 0.001; Aβ 42/α-Syn: AUC = 0.710; C.I. 95%(0.572–0.848); *p* = 0.004) in comparison to Ng and α-Syn, respectively. Furthermore, when comparing the Aβ 42/Ng and Aβ 42/α-Syn ratios with Aβ 42/40, we did not find any statistically significant difference (Table 3).

### 2.4. Biomarkers Related to Synaptic Dysfunctions and AD-Core Biomarkers in AD Subgroups

We also assessed the relationship of CSF proteins and cognitive impairment in AD. With this aim, we stratified AD patients into three groups [59]: MCI due to AD, mild AD and moderate AD.

As shown in Table 4, we found statistically significant differences between MCI and moderate AD, with lower Aβ 42/40 ratios (*p* = 0.014) and Aβ 42/Ng ratios (*p* = 0.006), and higher pTau (*p* = 0.026) in MCI than moderate AD. We found no differences between MCI and mild AD, or between mild and moderate AD.

### 2.5. Effect of ApoE4 Genotype on CSF Ng and α-Syn Concentrations in AD Group

To assess the influence of the ApoE4 genotype on CSF Ng and α-Syn, we stratified AD patients into two subgroups: ApoE 4(+) (i.e., patients carrying the ApoE ε4 allele, both in homo- and heterozygosis; *n* = 19) and ApoE 4(−) (i.e., patients carrying other ApoE alleles; *n* = 50). Patients with ApoE4(+) showed statistically significantly higher CSF Ng levels than ApoE4 (−) (*p* < 0.001), but the same difference was not found for α-Syn (*p* = 0.546) (Figure 3).

### 2.6. Correlations of CSF Levels of Synaptic-Related Biomarkers with Clinical Features in AD Group

To investigate the relationship of the CSF Ng and α-Syn levels, and Aβ-42/Ng and Aβ-42/α-Syn ratios with demographic and clinical features in AD, we performed Spearman’s correlation analyses. We found a significant positive correlation of the CSF Ng levels with the age at time of LP (rho = 0.250; *p* = 0.002) and age at onset (rho = 0.187; *p* = 0.008), and a significant negative relationship with MMSE scores (rho = −0.215; *p* = 0.037), while CSF α-Syn, although showing a positive correlation with the age at time of LP (rho = 0.280; *p* < 0.001) and with the age at onset (rho = 0.236; *p* = 0.003), did not display a significant correlation with MMSE. Moreover, Aβ-42/Ng showed a stronger positive correlation with MMSE (rho = 0.447; *p* < 0.001) than Aβ-42/α-synuclein (rho = 0.368; *p* = 0.003) (Table 5).

### 2.7. Relationship of Synaptic-Related Biomarkers with Clinical Features and AD-Core Biomarkers

To investigate the relationships of synaptic-related biomarkers (i.e., Ng, α-Syn, the Aβ 42/Ng ratio, and the Aβ 42/α-Syn ratio), non-parametric variables were log10-normalized, and then, we performed univariate linear regression analysis to investigate the influence of clinical features and AD-core biomarkers on their values (Appendix A).

The predicted value of Ng was higher for patients with MMSE scores less than 24 points (MMSE < 24); for pathological values of the Aβ 42/40 ratio (i.e., <0.066), pTau (i.e., >61 pg/mL), and tTau (i.e., >416 pg/mL); and in the presence of the ApoE ε4 (ApoE4) genotype. We considered the pathological cut-off values for the AD-core biomarkers suggested by the manufacturer. For α-Syn, the estimated value was higher for pathological values of the Aβ 42/40 ratio and pTau. Considering Aβ 42/Ng, we expected lower values with MMSE < 24, in the presence of the AD-related PET pattern and of the ApoE4 genotype, and with pathological values of Aβ 42, the Aβ 42/40 ratio, pTau and tTau. Finally, the predicted value of the Aβ 42/α-Syn ratio was lower with MMSE < 24, in the presence of the PET AD-related pattern, and with pathological values of Aβ 42, the Aβ 42/40 ratio, pTau and tTau.

Then, hierarchical linear regression analyses were performed to evaluate how AD-core biomarkers (i.e., Aβ 42/40, pTau and Tau) influenced the variance of synaptic-related biomarkers (Table 6).

For Ng and α-Syn, considering, in the base model, the age at LP, gender, education and pathological values of the MMSE, we showed that the contribution of the base model explained 0.4% and 29% of the variance in the Ng and α-Syn levels, respectively. The addition of pathological values of the Aβ42/40 ratio explained, respectively, 2.8% and 15.6% of the variance of the Ng and α-Syn levels above and beyond the base model. Considering the Aβ-42/Ng ratio, the base model (Aβ-42/Ng ~ age at LP + gender + education + pathological MMSE + AD-PET + ApoE4) explained 62% of the variance in its value. Model 1 included a pathological Aβ42/40 ratio as a predictor explaining an additional 17.1% of the variance in the Aβ-42/Ng ratio value above and beyond the base model. The addition of pathological values of pTau explained an additional 3% of the variance above and beyond Model 1. However, the addition of pathological levels of tTau did not contribute to elaborating a model with an increased variance above and beyond Model 2. When we considered Aβ-42/α-Syn, the base model (Aβ-42/α-Syn ~ age at LP + gender + education + pathological MMSE + AD-PET) explained 49.4% of the variance in its value. In Model 1, the addition of a pathological Aβ42/40 ratio as a predictor explained an additional 22% of the variance above and beyond the base model. Moreover, the addition of pathological levels of pTau and tTau did not contribute to elaborating a model with an increased variance above and beyond Model 1.

## 3. Discussion

The purpose of this study was to investigate the role of the synaptic proteins Ng and α-Syn in the CSF as potential additional biomarkers for the diagnosis of AD. We performed a retrospective observational study, in which Ng and α-Syn were analyzed in CSF from patients affected by AD, neurodegenerative disorders (*n*-AD), and other non-neurodegenerative neurological disorders (*n*-ND).

According to previous literature [6], we found higher CSF concentrations of Ng and α-Syn in comparison with the two control groups. However, when we considered the ability of Ng and α-Syn to discriminate AD from *n*-ND by ROC curve analyses, we obtained high sensitivity and low specificity for both proteins. When we analyzed the ROC curves to differentiate AD from *n*-AD, we observed a fair discriminatory capacity for Ng, with a specificity of 78% and a sensitivity of 68%, while no significant results were obtained for α-Syn. Then, considering previous data reported in the literature [56], we investigated the role of the CSF Aβ42/Ng and Aβ42/α-Syn ratios in improving the differential diagnosis of AD and *n*-AD, such as Aβ42/40 did. We found significantly lower values of Aβ42/Ng and Aβ42/α-Syn in AD than in *n*-AD. Moreover, we found that, similarly to Aβ 42, the Aβ 42/Ng and Aβ 42/α-Syn ratios performed better than Ng and α-Syn alone.

When we stratified AD patients into three subgroups according to MMSE scores, we found significant differences between MCI and moderate AD when we evaluated the Aβ 42/40 ratio, pTau and the Aβ 42/Ng ratio. Interestingly, the Aβ 42/Ng ratio was able to discriminate MCI from moderate AD. Although we did not find significant differences in CSF Ng concentrations between MCI and mild AD, as previously reported [35], probably because of the small sample, Aβ42/Ng was found to be higher in MCI than moderate AD, and the same was true for the Aβ 42/40 ratio, while we found no difference for α-Syn and Aβ 42/α-Syn. Our results could reflect the progression of synaptic pathology along the disease.

We also stratified AD patients according to the presence/absence of the ApoE ε4 allele and found higher levels of CSF Ng in ApoE4(+) carriers (Figure 3). An association between the ApoE genotype and synaptic loss in AD was widely demonstrated [60,61]. However, results regarding the relationship between ApoE and CSF Ng are contrasting [54,56,62]. Fan and colleagues found higher levels of CSF Ng in ApoE 4(+), suggesting a role for the relationship between ApoE and Ng in the pathophysiology of AD [54].

Upon analyzing the relationship of synaptic-related biomarkers with the demographic and clinical features of AD patients, we found that both CSF Ng and α-Syn were positively correlated with the age of the patients at onset, and at the time of diagnosis, Ng correlated negatively with the MMSE, while Aβ 42/Ng and Aβ 42/α-Syn positively correlated with the MMSE. These results confirmed previous findings in which Ng was involved in cognitive deterioration and considered a “cognitive biomarker” [50].

To estimate the contribution of clinical features and AD-core biomarkers in modulating the levels of synaptic-related biomarkers, we firstly performed a univariate regression analysis, separately considering the contribution of Ng, α-Syn, Aβ 42/Ng and Aβ 42/α–Syn. For this purpose, we stratified patients on the basis of pathological MMSE scores (i.e., <24); pathological values of the Aβ 42/40 ratio (i.e., <0.066), pTau (i.e., >61 pg/mL) and tTau (i.e., >416 pg/mL); and the presence/absence of the ApoE4 genotype (both in homo- and in heterozygosis) and AD-related PET pattern. Interestingly, we found that both the Aβ 42/Ng and Aβ 42/α-Syn values are influenced by the presence of the AD-related PET pattern; moreover, the Ng concentration and Aβ 42/Ng value were influenced by the presence of the ApoE4 genotype and pathological values of AD-core biomarkers. Then, we analyzed the influences of different AD-core biomarkers on the levels of synaptic-related biomarkers, elaborating different models of prediction by hierarchical linear regression. Considering a basal model in which the age at LP, education, gender, MMSE, AD-related PET, and presence of ApoE4 were simultaneously considered, we obtained a prediction model in which the serial addition of pathological values of Aβ 42/40 and pTau was able to explain 82.1% of the variance in the Aβ 42/Ng ratio value.

Despite a specific role for Ng in AD pathogenesis still being unclear, its location in the postsynaptic dendritic spines of the hippocampal neurons [63] makes this molecule a potential biomarker of synaptic dysfunction in AD. Indeed, most authors found that the CSF Ng concentration is higher in AD than in healthy controls, but also in MCI due to AD, in preclinical AD and even in elderly people [35,64,65,66,67]. It positively correlates with hippocampal atrophy [37] and with CSF tTau and pTau, and negatively correlates with cognitive performance, Aβ 42 and Aβ 42/40 [29,35,65]. Ng also seems to predict the progression from MCI to AD [34,37]. However, Ng was found to also be associated with different neurodegenerative conditions such as PD and CJD [35,40], and it is not able to distinguish AD from LBD and FTD [34]. Moreover, in a postmortem study, the Ng in the CSF was not correlated with the Ng load in the brain tissue.

In this study, we wanted to analyze the diagnostic performance of Aβ 42/Ng, which has previously been studied as an index of synaptic dysfunction [28,56,57,58]. Physiologically, amyloid plays a role in synaptic function, while Aβ 42 is toxic to neurons at the presynaptic level. That is why Sancesario recently tested Aβ42/Ng as an index of synaptic failure in the cognitive dysfunction of PD patients [58], showing that both Ng and Aβ42/Ng correlated with the MMSE and were able to discriminate patients with cognitive decline, with the Aβ42/Ng ratio showing a better performance even when corrected for age and sex. Our study showed that, even in AD patients, Aβ42/Ng performed better than Ng alone in discriminating AD from controls, with higher sensitivity and specificity and a better correlation with the core AD biomarkers and with cognition, measured with MMSE scores. Moreover, Aβ42/Ng showed a slightly better AUC than Aβ42/Aβ40 in differentiating AD from *n*-AD. Our findings confirm and reinforce previous results obtained by Janelidze, who demonstrated a better diagnostic performance of Aβ42/Ng compared to Ng, but our results show stronger diagnostic value for Aβ42/Ng compared to Aβ42/Aβ40. Therefore, our results support the role of the Aβ 42/Ng ratio as a diagnostic biomarker and lay the groundwork for further studies, which will investigate its role as a prognostic factor.

With a similar reasoning, we also tested Aβ42/α-Syn. α-Syn has already been associated with AD pathology, and aggregates were found in patients affected by AD [29,37,44,45,49,57,68,69], but the ratio has never been tested before. Our results showed that Aβ42/α-Syn performed better than α-Syn alone, but worse than Aβ42/Ng in terms of the correlation with clinical features and diagnostic performance.

Altogether, our findings support the involvement of synaptic dysfunction in AD pathogenesis. Indeed, synaptic dysfunction is considered one of the early events that participates in the etiopathogenesis of AD and may be involved in the rate of the progression of the disease in terms of the impairment of cognitive functions. The mechanisms leading to the synaptic dysfunctions, at both the pre- and postsynaptic levels, include Aβ peptides and Tau proteins, which participate in the incredibly early stages, before the formation of amyloid plaques and neurofibrillary tangles, respectively. In fact, in pathological conditions, Aβ peptides can form soluble oligomers, which accumulate at the presynaptic level, impairing axonal transport, synaptic vesicle cycling, and the release of neurotransmitters. Similar effects are observed in the case of the progressive phosphorylation of Tau proteins [26,27].

## 4. Materials and Methods

### 4.1. Study Population

We present a retrospective observational study, which included sixty-nine patients affected by AD, fifty patients affected by neurodegenerative diseases (*n*-AD), and ninety-eight patients affected by non-neurodegenerative neurological disorders (*n*-ND), whose details are reported in Table 1, Appendix A.

The AD and *n*-AD patients were attending the Unit of Neurology, Department of Biomedicine, Neuroscience, and Advanced Diagnostic (Bi.N.D.—University of Palermo, Palermo, Italy), while the *n*-ND patients were attending the ALS Clinical Research Center (Bi.N.D.—University of Palermo, Palermo, Italy). We considered patients from 2018 to 2022.

The AD and *n*-AD patients underwent complete medical history analysis, clinical examinations, neuropsychological testing, and structural brain (MRI) and metabolic (FDG-PET) neuroimaging. Furthermore, the ApoE genotype was investigated for the AD and *n*-AD patients.

The AD patients received the diagnosis of “Probable AD dementia with high of evidence of AD pathophysiological process” or “MCI due to AD—High likelihood” according to the published criteria (McKhann et al. [11] and Albert et al. [12]).

All the patients gave their written informed consent, which contained the statement “the biological material may also be used for research purposes”, and all the procedures were conducted in accordance with the Declaration of Helsinki and its amendments. The Ethics Committee of the University Hospital of Palermo approved the study protocol.

### 4.2. Neuropsychological Assessment

Cognitive deficits were measured by the Mini Mental State Examination (MMSE). This is a widely used test for cognitive impairment in clinical practice because of its reproducibility and ease of administration. The MMSE is heavily influenced by age and education and provides information about the degree of general cognitive impairment: normal (24–30), mild (19–23), moderate (18–10) and severe (≤9) [70].

### 4.3. FDG-PET

The AD and *n*-AD patients underwent FDG-PET scans during the diagnostic work-up. Brain areas were evaluated on the basis of their metabolic statuses. We define as an “AD-related PET” pattern the presence of a hypometabolism spatial pattern in areas that are generally involved in AD (i.e., the hippocampus, medial temporal cortex, lateral temporal cortex, superior and inferior parietal gyri, posterior cingulate and precuneus) [71].

### 4.4. CSF Sampling, Processing and Analyses

Lumbar puncture (LP) was performed in fasted patients between 8:00 a.m. and 10:00 a.m. during the diagnostic work-up. CSF samples were collected in polypropylene tubes, centrifuged at 300× *g* for 5 min to remove cell debris and blood contamination, aliquoted in propylene tubes, and stored at −80 °C until analysis, according to international consensus protocols [72].

The CSF Ng P75 and α-Syn levels were measured using commercially available ELISA kits, according to the manufacturer’s instructions [30]. The CSF Aβ42, Aβ40, pTau and tTau levels, which represent AD-core biomarkers, were measured using the chemiluminescence enzyme immunoassay (CLEIA), using commercially available kits [4], according to the manufacturer’s instructions. The analyses of AD-core biomarkers were performed only for AD and *n*-AD patients.

### 4.5. Statistical Analyses

All the statistical analyses were performed using SIGMAPLOT 12.0 (Systat Software Inc., San Jose, CA, USA), SPSS version 25 (SPSS Inc., Chicago, IL, USA), and R Language v.4.1.2 (R Foundation for Statistical Computing, Vienna, Austria) software packages. 

The distribution of the data was assessed with the Shapiro–Wilk method. Normally distributed variables are expressed as the mean ± standard deviation (sd); skewed variables, as median and IQR values; and categorical variables, as absolute and relative frequencies.

For skewed variables, we used the Mann–Whitney U test and Kruskal–Wallis one-way ANOVA on ranks, while the differences in normal variables were analyzed through the one-way ANOVA. In the case of statistical significance, Dunn’s post hoc test was used for pairwise comparisons, and *p*-values were adjusted for multiplicity by the Holm–Sidak method. The differences between groups for categorical variables were estimated using the chi-square test or Fisher’s exact test, as appropriate.

The discriminatory ability of the studied biomarkers to correctly allocate the participants to different diagnostic groups was evaluated by receiver operating characteristic curve (ROC) analysis and is reported as the area under the curve (AUC) and 95% confidence interval (C.I.). The best statistical threshold for Ng, α-Syn, the Aβ 42/Ng ratio and the Aβ 42/α-Syn ratio, in the prediction of AD diagnosis, was estimated using the Youden method, selecting the threshold at which the quantity Youden’s index = ([sensitivity + specificity] − 1) was maximized. The discriminatory ability of biomarkers on the basis of the AUC was classified as follows: “excellent” (AUC 0.90–1.00), “good” (AUC 0.80–0.89), “fair” (AUC 0.70–0.79), “poor” (AUC 0.60–0.69) or “fail”/no discriminatory capacity (AUC 0.50–0.59). Differences in the area under the ROC curve (AUC) for two ROC curves were compared using the bootstrap method [73].

The statistical correlation between continuous variables was evaluated by Pearson’s analyses, while non-parametric data were analyzed using Spearman’s rho.

The associations of synaptic-related biomarkers with clinical features and AD-core biomarkers were then specifically assessed by univariate linear regression, as a simple model. For this purpose, non-parametric data were log-10 transformed. Then, we used competitive models leveraging a hierarchical linear regression approach to evaluate the contribution to the variance explained by each significant predictor from the independent analyses. The model selection was aided by Akaike information criterion (AIC) and Bayesian information criterion (BIC) calculations.

A *p*-value less than 0.05 was considered statistically significant.

## 5. Conclusions

Our research strengthens the role of the synaptic proteins Ng and α-Syn as biomarkers of synaptic dysfunction, and highlights the potential of the Aβ42/Ng ratio as a reliable biomarker for the early diagnosis of AD. Synaptic pathology is increasingly considered a central feature in the pathogenesis of AD, and synaptic proteins represent an easily measurable feature in AD. AD biomarkers are very specific and sensitive for early diagnosis, but in the “real world”, comorbidities and atypical presentation may limit their interpretation. Larger longitudinal studies are needed to validate the Aβ42/Ng ratio as a biomarker to be added to the panel for AD diagnosis.

## Figures and Tables

**Figure 1 ijms-23-10831-f001:**
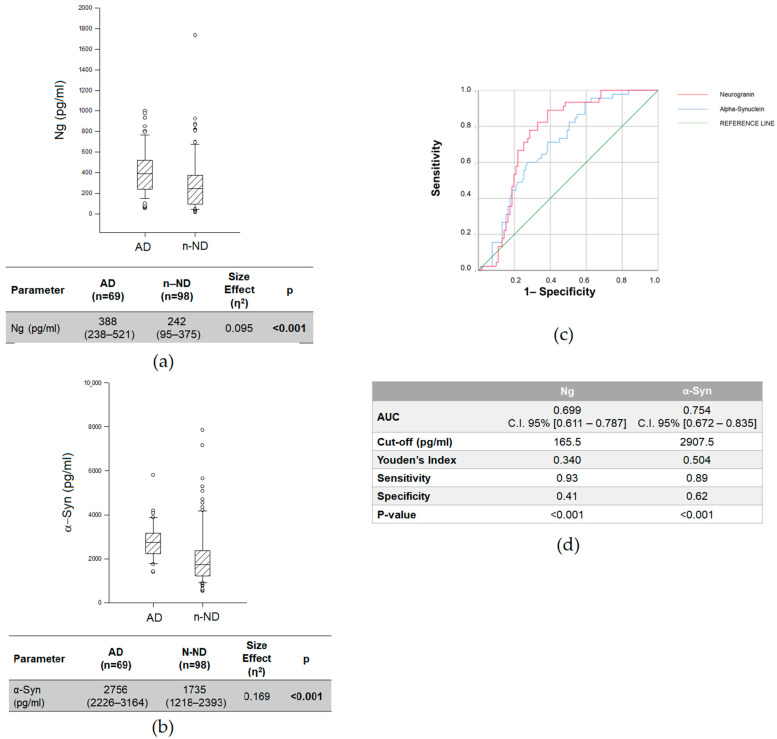
CSF Ng (**a**) and α-Syn (**b**) levels in AD and *n*-ND patients. Data are expressed as medians with interquartile ranges (IQRs). We used the Kruskal–Wallis test with Dunn’s post hoc test. *p* < 0.05 was considered statistically significant and is indicated with bold font. (**c**) Receiver operating characteristic (ROC) analyses and (**d**) area under curve (AUC) of CSF Ng and α-Syn in AD vs. *n*-ND.

**Figure 2 ijms-23-10831-f002:**
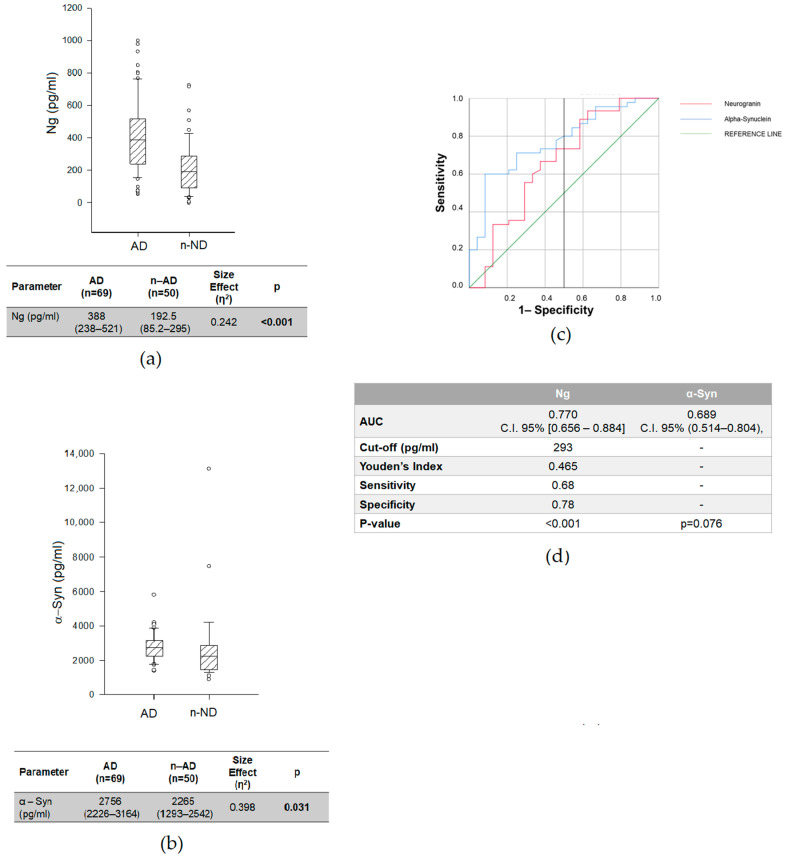
CSF Ng (**a**) and α-Syn (**b**) levels in AD and *n*-AD patients. Data are expressed as medians with interquartile ranges (IQRs). We used the Kruskal–Wallis test with Dunn’s post hoc test. *p* < 0.05 was considered statistically significant and is indicated with bold font. (**c**) Receiver operating characteristic (ROC) analyses and (**d**) area under curve (AUC) of CSF Ng and α-Syn in AD vs. *n*-AD.

**Figure 3 ijms-23-10831-f003:**
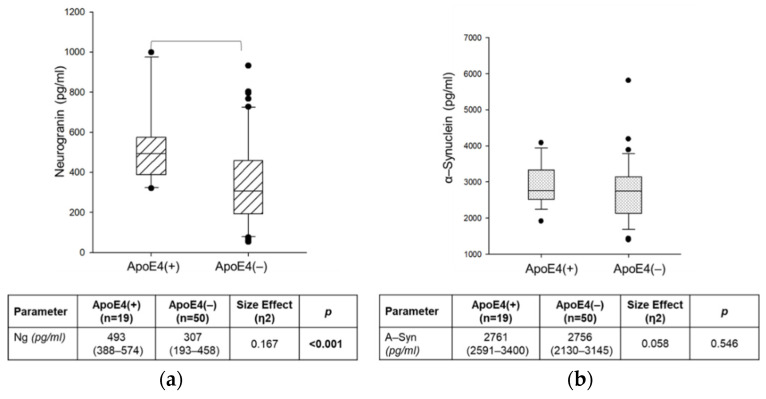
CSF neurogranin (**a**) and α-synuclein (**b**) levels in AD patients subgrouped based on ApoE ε4 genotypes: ApoE4(+) and ApoE4(−). Data are expressed as medians with interquartile ranges (IQRs) and were analyzed by the Mann–Whitney U test, also indicating the size effect (η^2^). *p*-values were corrected for multiplicity by the Holm–Sidak method and are indicated with bold font.

**Table 1 ijms-23-10831-t001:** Demographic, clinical and CSF features of patients with Alzheimer’s disease (AD), patients with other neurodegenerative disorders (*n*-AD) and patients with non-neurodegenerative disorders (*n*-ND). Data are expressed as medians with interquartile ranges (IQRs).

Variables	AD (*n* = 69)	*n*-AD (*n* = 50)	*n*-ND (*n* = 98)	*p*
**Demographic and clinical features**				
Age at LP ^£^ (*years*)	72 (67.5–76)	69 (59.5–73)	63 (47–71)	**<0.001****^A^**^,^**^,^***
Age at onset (*years*)	67 (62–71.5)	63 (55–71)	59 (44–68)	**<0.001****^A^**^,^**
Gender (*M/F*)	1.26	1.42	1.67	0.727 ^B^
Education (*years*)	8 (5–13)	8 (5–8)	8 (5–13)	0.156 ^A^
Diagnostic delay (*years*)	4 (3–6)	3 (2–5.5)	2 (1–4)	**<0.001 ^A^**^,^**^,^***
MMSE scores (*raw*)	23 (15–26)	21 (17–28)	29 (28–30)	**0.012 ^A^**^,^**^,^ ***
**CSF biochemical features**				
Protein (mg/dL)	39 (31.8–48.8)	42.7 (31.1–56.4)	45 (31–57)	0.337 ^A^
Glucose (mg/dL)	59.3 (56–64)	60 (53–67)	62 (55–71)	0.346 ^A^
Cells (*n*/mmc)	0.8 (0.6–2)	0.8 (0.6–2)	1.2 (0.8–2.4)	0.058 ^A^

£ Lumbar puncture; A Kruskal–Wallis one-way analysis of variance on ranks; B chi-square; bold font indicates statistical significance (*p* < 0.05); post hoc Dunn’s method: ** AD vs. *n*-ND; *** *n*-AD vs. *n*-ND.

**Table 2 ijms-23-10831-t002:** Aβ 42/40, Aβ 42/Ng and Aβ 42/α-Syn values in AD and *n*-AD patients.

	AD (*n* = 69)	Non-AD (*n* = 50)	Size Effect (η^2^)	*p*
Aβ 42/40 Ratio	0.051 (0.040–0.079)	0.098 (0.072–0.102)	0.233	**<0.001**
Aβ 42/Neurogranin	1.443 (0.822–2.861)	4.170 (2.222–5.613)	0.280	**<0.001**
Aβ 42/α-Synuclein	0.231 (0.149–0.383)	0.425 (0.298–0.480)	0.131	**0.001**

Data are expressed as medians with interquartile ranges (IQRs) and were analyzed by Mann–Whitney U tests, also indicating the size effect (η^2^). *p*-values were corrected for multiplicity by the Holm–Sidak method and are indicated with bold font.

**Table 3 ijms-23-10831-t003:** ROC analyses of the CSF biomarkers.

Variables	AUC	C.I. 95%	*p*	AUC Difference Versus Aβ42/Aβ40 (*p*-Value)
Aβ 42/40 Ratio	0.802	0.672–0.933	**0.002**	
Neurogranin	0.768	0.682–0.853	**0.004**	
A-Synuclein	0.689	0.514–0.804	0.076	
Aβ 42/Neurogranin	0.814	0.684–0.944	**0.001**	0.005 (0.943)
Aβ 42/α-Synuclein	0.710	0.572–0.848	**0.004**	−0.066 (0.420)

Bold font indicates that the result is statistically significant (*p* < 0.05).

**Table 4 ijms-23-10831-t004:** CSF levels of AD-related biomarkers (i.e., Aβ 42, pTau, tTau, Aβ 42/40, Aβ 42/pTau and Aβ 42/tTau) and biomarkers related to synaptic dysfunction (i.e., neurogranin, α-synuclein, Aβ 42/neurogranin and Aβ 42/α-synuclein) in AD subgroups (i.e., MCI, mild AD and moderate AD).

Variables	MCI (*n* = 28)	Mild AD (*n* = 14)	Moderate AD (*n* = 27)	Size Effect (η^2^)	*p*
Aβ 42 (pg/mL)	677 (495–971)	599 (421–725)	521 (413–674)	0.041	0.093
Aβ 42/40 Ratio	0.060 (0.043–0.102)	0.044 (0.036–0.059)	0.046 (0.038–0.052)	0.100	**0.014 ***
pTau (pg/mL)	65.0 (32.4–88.0)	79.2 (44.8–121.7)	95.8 (65.1–154.6)	0.080	**0.026 ***
tTau (pg/mL)	550 (284–645)	541 (412–810)	678 (495–896)	0.053	0.065
Neurogranin (pg/mL)	338 (235–492)	417 (217–526)	388 (260–526)	0.000	0.852
α-Synuclein (pg/mL)	2754 (2135–3143)	3082 (2266–3883)	2556 (2377–3124)	0.000	0.434
Aβ 42/Neurogranin	2.055 (1.015–3.639)	1.221 (0.838–2.522)	0.865 (0.400–1.578)	0.126	**0.006 ***
Aβ 42/α-Synuclein	0.313 (0.187–0.423)	0.162 (0.144–0.280)	0.202 (0.141–0.249)	0.051	0.068

Data are expressed as medians with interquartile ranges (IQRs) and were analyzed by Kruskal–Wallis ANOVA on ranks with the post hoc Dunn’s method, also considering the size effect (η^2^). Statistically significant *p*-values (*p* < 0.05) are indicated with bold font. * Post hoc Dunn’s method: MCI vs. moderate AD.

**Table 5 ijms-23-10831-t005:** Spearman’s correlations of CSF neurogranin and α-synuclein levels, and Aβ-42/neurogranin and Aβ-42/α-synuclein ratios with demographic and clinical features of AD patients.

Variables	Neurogranin	α-Synuclein	Aβ-42/Neurogranin	Aβ-42/α-Synuclein
Age at LP	0.250; ***p* = 0.002**	0.280; ***p* < 0.001**	−0.107; *p* = 0.393	0.397; ***p* < 0.001**
Age at Onset	0.187; ***p* = 0.008**	0.236; ***p* = 0.003**	0.063; *p* = 0.594	0.321; ***p* < 0.001**
Education	−0.157; *p* = 0.157	−0.113; *p* = 0.360	0.044; *p* = 0.723	−0.043; *p* = 0.731
MMSE	−0.215; ***p* = 0.037**	−0.174; *p* = 0.170	0.427; ***p* < 0.001**	0.368; ***p* = 0.003**

**Table 6 ijms-23-10831-t006:** Competitive hierarchical linear regression results.

Model	Formula	AIC	BIC	R^2^	Adjusted R^2^	ΔR^2^
Ng						
Base	~Age at LP + Gender + Education + MMSE	263.919	270.209	0.511	0.417	0.004
1	~Base + Aβ42/40	259.499	267.048	0.618	0.522	0.028
α-Syn						
Base	~Age at LP + Gender + Education + MMSE	275.412	280.635	0.290	0.113	0.290
1	~Base + Aβ42/40	272.220	278.487	0.446	0.261	0.156
Aβ 42/Ng						
Base	~Age at LP + Gender + Education + MMSE + AD-PET + ApoE4	17.856	26.663	0.620	0.500	0.620
1	~Base + Aβ42/40	4.280	14.344	0.791	0.710	0.171
2	~Base + Aβ42/40 + pTau	2.128	13.451	0.822	0.738	0.031
Aβ 42/α-Syn						
Base	~Age at LP + Gender + Education + MMSE + AD-PET	−209.986	−204.011	0.494	0.313	0.494
1	~Base + Aβ42/40	−219.376	−212.406	0.714	0.582	0.220

ΔR^2^ = change in R ^2^ from previous nested model. Akaike information criterion (AIC) and Bayesian information criterion (BIC) calculations derived as follows: AIC = 2K − 2 *ln* (L), where K = number of model parameters, and *ln* (L) = model log-likelihood. BIC = (RSS + log(n)d σˆ2)/*n*, where RSS = residual sum of squares. *n* = Total observations. d = Number of predictors. σˆ = Estimate of variance of the error associated with each response measurement.

## Data Availability

The data presented in this study are available on request from the corresponding author. The data are not publicly available due to current privacy laws.

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
