# Peer review of "Biomarkers Related to Synaptic Dysfunction to Discriminate Alzheimer’s Disease from Other Neurological Disorders"

_ijms, 2022, doi:10.3390/ijms231810831_

Round 1
Reviewer 1 Report
In this study, Piccoli et al investigate the role of Neurogranin and presynaptic α-synuclein as potential biomarkers to discriminate Alzheimer´s disease from other neurodegenerative disorders. For this purpose, they measure the CSF levels of these proteins and perform a thorough statistical analysis, comparing the results obtained between Alzheimer´s patients, patients affected by other neurodegenerative disorders and individuals not affected by a neurodegenerative disorder. The main conclusion of the study is that Neurogranin is a potential biomarker for Alzheimer´s disease and that the Aβ 42/Ng ratio can be used to discriminate Alzheimer´s disease from other neurodegenerative diseases.
This is a straightforward study in which a strong statistical analysis is presented, including interesting variables that provide more information about different neurological conditions. However, the manuscript presents some major issues that should be addressed.
Lane 26: I would recommend using a different abbreviation for no-Degenerative Disorders, considering that n-ND is very similar to no-AD Neurodegenerative Disorders (nAD-ND) and this makes the understanding of the results complicated.
In general, the introduction needs to be improved since many concepts that are presented in the results are not reviewed in the introduction. This makes the manuscript very difficult to understand for people who are not experts in neurodegenerative disorders. These include:
-Mini Mental State Examination (MMSE) scores: Why is this performed to patients? Which information provides? What is this Examination about?
-Oligoclonal bands: What are they? What is their importance in these diseases?
- Aβ 42/Ng ratio: What kind of information this ratio provides?
-pTau and tTau: What are these markers? What is the difference between them? What information can we obtain from measuring their levels and what is the implication for the disease?
-ApoE ε4 genotype: What is this? What are the consequences for patients?
I would also suggest including a short paragraph at the beginning of the results instead of starting directly with Table 1. I do not understand some of the information provided on this table. It seems to be duplicated for some reason. In addition, the symbols that are used at the foot do not coincide with those used on the table. It states that Kruskal Wallis test with post-hoc pairwise test was performed, but I only see a single column for p value, and I do not understand which statistical analysis was used to obtain that value of p. At the bottom it states that *** was used for Chi square test, but I cannot find this symbol on the table.
In Figure 1 there are several outlier values. Could it be possible that significant results were obtained because of these measures? No explanation for these results is given in the discussion. The discussion needs to be more thorough, comparing the results obtained here with those obtained in other studies in which Ng levels have been measured in humans and animals affected with neurodegenerative diseases.
In general, the results are presented in a very technical way, with no brief explanation of the data obtained, which makes the understanding of this section difficult.
In the conclusions, it is stated that CSF Ng is a potential biomarker for the diagnosis of AD. However, I understood that the main conclusion of the study was that Aβ 42/Ng ratio was better at discriminating AD from other disorders than Ng levels, so I believe this is a more important conclusion that is not included in this section. CSF levels of Ng have been evaluated in several studies before, not only in Alzheimer´s, but also in other neurodegenerative diseases.
Some paragraphs need to be rephrased since the understanding is complicated: Lanes 59-63, lanes 272-275, lanes 294-299.
There are some minor errors within the text:
-Lane 29 “Significant AD” . “significant” seems to be a typo
-Lane 92: Summarizes instead of summaries.
-Lane 59: Arises instead of arise.
Author Response
We thank the reviewers for the suggestions. We will respond one by one.
Line 26: I would recommend using a different abbreviation for no-Degenerative
Disorders…
We modified the abbreviation nAD-ND with n-AD in the text
- In general, the introduction needs to be improved
We added more concepts and references to clarify the meaning of the manuscript:
1. We briefly explained MMSE and its utility
2. We perform oligoclonal bands in our routine analyses, but they have not a role
in present work. We excluded them from the text
3. We discussed the Aβ 42/Ng ratio in the introduction
4. We described pTau and tTau, their difference and their role in AD pathogenesis
5. We added a brief paragraph about APOE genotype and its role as risk factor in
AD. We also briefly reported the possible interaction of APOE e4 with
Neurogranin and Synuclein
I would also suggest including a short paragraph at the beginning of the results instead of starting directly with Table 1
A short paragraph has been added to the start of the results as requested. We
also modified table 1, making it more readable (yes it was duplicated by
mistake)
In Figure 1 there are several outlier values. Could it be possible that significant results were obtained because of these measures?
We conducted our analyses using median values, not means, to ensure that they were not influenced by outliers. However, we re-analyzed our data excluding outliers and the significance is confirmed. We have decided to leave our original
analyses in the manuscript because we consider them stronger.
In general, the results are presented in a very technical way, with no brief explanation of the data obtained, which makes the understanding of this section difficult.
We rephrased the entire paragraph, adding additional results.
In the conclusions, it is stated that CSF Ng is a potential biomarker for the diagnosis of AD. However, I understood that the main conclusion of the study was that Aβ 42/Ng ratio was better at discriminating AD from other disorders than Ng levels, so I believe this is a more important conclusion that is not included in this section.
We modified the discussion, stressing the importance of Aβ 42/Ng ratio in our work, and the same the conclusion.
Some paragraphs need to be rephrased since the understanding is complicated: Lanes 59-63, lanes 272-275, lanes 294-299.
We have reworded the paragraph and hope that it is now more readable.
Reviewer 2 Report
ID: ijms-1859291
Title: Biomarkers related to synaptic dysfunction to discriminate Alzheimer’s disease from other neurological disorders.
The authors engaged in a scientific effort along the clinical validation and qualification of CSF biomarkers for Alzheimer’s clinical practice and pharmacological trials.
While such an endeavor is valuable from a theoretical standpoint, the article suffers from several methodological limitations that hinder any attempt to reach solid clinical conclusions.
Without an extensive revision process that stems from a conceptual revision of the scientific questions, goes through a reconsideration of specific steps of the statistical workplan, and concludes with further elaboration of the Discussion, any novelty inherent to the study does not really come out.
Major points
1) The Introduction suffers from several weak points, including an overview of the existing literature about Neurogranin and alpha-synucleins. While the Introduction provides information about the neurochemistry of the candidate biomarkers (though the present article does not set itself as a discovery study but rather a diagnostic test validation study), it fails to devise a concise state-of-the-art across the AD continuum. Such a shortcoming reflects a poorly developed Discussion (see comment below).
The current version of the article misses out on several key papers in the field of AD bodily fluids biomarkers, with particular regard to Ng and a-syn (e.g., Tarawneh R, et al. PMID: 27018940; Lista S, et al. PMID: 28731449; Casaletto KB, et al. PMID: 28939668; Vergallo A, et al.; Wang H, et al. PMID: 29376878; among others).
Without a comprehensive introduction of the current landscape, it is challenging to grasp any original clinical value of the study.
2) The authors focus on the diagnostic value of Ng to distinguish AD from other conditions; though several critical concerns stand out after reading the piece:
2a. The authors included MCI and mild dementia AD individuals in the study, merging them into a single population; however, they did not provide results stratified by disease stage, which, makes enormous difference from a biological and perspective therapeutic (i.e., the molecularly targeted therapies the authors bring up in the Discussion).
2b. No association was found (or at least reported) for Ng and a-syn with AD validated biomarkers; in this sense, the authors claim that Ng may be integrated into the AD diagnostic biomarker panel without supplying convincing argumentation (both study-wise and literature-based) about the statistical and biological relationship between this candidate biomarker and AD amyloid-beta or tau pathology.
The same reasoning should be developed for a-syn and AD biomarkers (see, for instance but not exclusively, the previously reported a-syn / p-tau181 mismatch - Wang H, et al. PMID: 29376878).
2c. The authors do not provide exhaustive evidence (results) about the adjunctive value of Ng to the established AD biomarker panel diagnostic performance. The AUC values displayed in ROC curves does not show whether adding Ng to the AD biomarker panel would significantly raise accuracy. While it is not mandatory for your study that Ng increases the overall accuracy, the authors shall either avoid overstatements in the Discussion or run additional analysis and then better flesh out how they would employ Ng in clinical practice for diagnostic purposes.
3) It is unclear whether the authors carried out the comparison analyses at once and whether these were adjusted for multiple comparisons. Alongside supplying effect size estimation coefficients and degree of freedom (currently difficult to find through the reading), the authors should clarify the multiple comparisons issue. Taken together, these points are pivotal for weighing the article overall clinical value.
4) It is unclear why there is no extensive, literature-based Discussion about the APOE-wise results concerning Ng. As the authors acknowledge, APOE is the most significant genetic risk factor for AD (they shall clarify sporadic, late-onset AD); thus, it is a bit awkward that such a result is not well addressed with also an outlook on other potential use of Ng (prognosis? disease risk stratification? other?).
5) Some blurred points come off the piece as the reader gets through it. For instance, Table 1 - supposed to report the demographic and clinical biomarkers data - does not provide any data about Ng and a-syn (is this a misprint?). Finally, some sentences are a little verbose, (e.g., CSF Ng is considerable as diagnostic biomarker for AD,..) and may benefit from a revision for English.
Author Response
We thank the reviewer for the suggestions. We will respond one by one.
1) “The Introduction suffers from several weak points, including an overview of the existing literature about Neurogranin and alpha-synucleins”.
We modified the introduction as suggested by the reviewer, to include more detailed evidences
from the literature and limited the chemical description of the molecules.
2) “The authors focus on the diagnostic value of Ng to distinguish AD from other
conditions, though several critical concerns stand out after reading the piece:
a. The authors included MCI and mild dementia AD individuals in the study,
merging them into a single population; however, they did not provide results
stratified by disease stage, which, makes enormous difference from a biological
and perspective therapeutic (i.e., the molecularly targeted therapies the authors
bring up in the Discussion).
The reviewer proposed the stratification of our population according to the stage of the disease. This is a very intriguing suggestion. For this purpose, we carried out this analysis and reported the results in the text. However, we would like to stress that all our MCI patients are diagnosed with AD: all of them received a diagnosis of “MCI due to AD—High likelihood” according to Albert 2011.
b. No association was found (or at least reported) for Ng and a-syn with AD
validated biomarkers; in this sense, the authors claim that Ng may be integrated into the AD diagnostic biomarker panel without supplying convincing argumentation (both study-wise and literature-based) about the statistical and biological relationship between this candidate biomarker and AD amyloid-beta or tau pathology.
We performed linear regression analyses for Ng and Syn with AD-core biomarkers and found interesting correlations supporting our conclusions. In particular, we obtained intriguing results when we considered Ab 42/Ng and Ab 42/α-Syn. We reported results in the manuscript.
c. The authors do not provide exhaustive evidence (results) about the adjunctive
value of Ng to the established AD biomarker panel diagnostic performance. The AUC values displayed in ROC curves does not show whether adding Ng to the AD biomarker panel would significantly raise accuracy. While it is not
mandatory for your study that Ng increases the overall accuracy, the authors shall either avoid overstatements in the Discussion or run additional analysis and then better flesh out how they would employ Ng in clinical practice for diagnostic purposes.
Firstly, we performed ROC analyses for synaptic-related biomarkers (that also included Ab42/Ng and Ab 42/α-Syn) and AD-core proteins. We found that Ab 42/Ng and Ab 42/α-Syn displayed a better diagnostic performance than Ng and a-Syn. We added the results into the manuscript and reviewed the discussion.
3) It is unclear whether the authors carried out the comparison analyses at once and whether these were adjusted for multiple comparisons. Alongside supplying effect size estimation coefficients and degree of freedom (currently difficult to find through the reading), the authors should clarify the multiple comparisons issue. Taken together, these points are pivotal for weighing the article overall clinical value.
For this purpose, we adjusted p-value for multiplicity by Holm-Sidak method and introduce for each comparison the size effect. In case of multiple comparison, we used Dunn’s post-hoc method.
4) It is unclear why there is no extensive, literature-based Discussion about the APOE-wise results concerning Ng. As the authors acknowledge, APOE is the most significant genetic risk factor for AD (they shall clarify sporadic, late-onset AD); thus, it is a bit awkward that such a result is not well addressed with also an outlook on other potential use of Ng (prognosis? disease risk stratification? other?).
We introduced the role of APOE genotype in the Introduction and discussed APOE findings in the Discussion.
5) Some blurred points come off the piece as the reader gets through it. Finally, some sentences are a little verbose, (e.g., CSF Ng is considerable as diagnostic biomarker for AD, etc.) and may benefit from a revision for English.
We carefully reviewed the English language in our manuscript.
Round 2
Reviewer 1 Report
Thank you, the manuscript has been significantly improved, however, figure captions should be at the bottom of the figure.
Author Response
We thank the reviewer for the comment. We corrected the figures, putting the figure captions at the bottom of each figure. The corrections are available in the uploaded manuscript version.